# Antimicrobial Properties of Thermally Processed Oyster Shell Powder for Use as Calcium Supplement

**DOI:** 10.3390/foods14152579

**Published:** 2025-07-23

**Authors:** Sungmo Ahn, Soohwan Lee, Seokwon Lim

**Affiliations:** Department of Food Science & Biotechnology, Gachon University, Sujeong-gu, Seongnam-si 13120, Republic of Korea; dkstjdah58@gachon.ac.kr (S.A.); tnghks3193@gmail.com (S.L.)

**Keywords:** oyster shells, calcium, thermal processing, antimicrobial, bioavailability

## Abstract

Oyster shells, though rich in calcium, are mostly discarded and contribute to environmental issues. Developing calcium-based materials with antimicrobial functionality offers a promising solution. However, their low bioavailability limits their direct use, requiring processing to enhance their applicability. Therefore, this study aims to evaluate the physicochemical properties and antimicrobial activity of thermally processed pulverized oyster shells (TPOS) and citric acid-treated TPOS (TPOSc) compared with those of fibrous calcium carbonate (FCC) and coral-derived calcium product (CCP), which are used as reference materials. The solubility values were 0.7 mg/g for FCC, 0.5 mg/g for TPOS, 0.4 mg/g for TPOSc, and 0.05 mg/g for CCP. The average particle sizes were 476 (FCC), 1000 (TPOS and TPOSc), and 1981 nm (CCP). Scanning electron microscopy (SEM), energy-dispersive X-ray spectroscopy (EDS), and X-ray diffraction (XRD) analyses revealed calcium ion release and structural changes in TPOS and TPOSc. Antibacterial testing further confirmed that these samples exhibited significant antimicrobial activity. Furthermore, to assess their practical applicability, TPOS and TPOSc samples with antimicrobial properties were incorporated into rice cakes. All samples retained antimicrobial activity at 0.3 wt%, while higher concentrations led to deterioration in their textural properties. These findings support the potential of thermally processed oyster shell powders for food applications that require microbial control with minimal impact on product quality.

## 1. Introduction

Oysters, saltwater bivalve mollusks, had a global production of approximately 6.1 × 10^6^ tons as of 2019, with their production continuously increasing due to aquaculture [1]. The byproducts of oyster production, oyster shells, are utilized to a limited extent (approximately 10%) in applications such as animal feed, fertilizers, and water treatment adsorbents, while over 90% are discarded as waste. However, due to limited landfill space and the absence of proper disposal methods, discarded oyster shells are often left untreated in the environment, which contributes to serious marine pollution and ecological damage [2]. Waste shells have been reported to cause environmental issues such as strong odor generation during microbial decomposition and heavy metal leaching from shell piles exposed to weathering [3]. Consequently, there is an increasing demand for research and development on the industrial utilization of oyster shells, driven by both environmental and economic factors [4].

Several upcycling studies have explored the extraction of functional ingredients from residual proteins in oyster shells for the development of functional cosmetics and food additives [5,6,7,8]. However, since the primary component of oyster shells is calcium carbonate (CaCO_3_), most of the oyster shells—excluding the minor protein fraction—has been widely utilized as a dietary calcium source [1]. To improve the solubility and bioavailability of oyster shell-derived calcium, it is typically subjected to high-temperature treatment (above 700 °C), which converts calcium carbonate into calcium oxide (CaO), followed by hydration to form calcium hydroxide (Ca(OH)_2_). According to previous studies, calcium oxide (CaO) obtained through this transformation process has been reported to exhibit antimicrobial activity and demonstrate inhibitory effects against pathogenic bacteria such as *Staphylococcus aureus* (*S. aureus*) and *Escherichia coli* (*E. coli*) [9,10,11,12]. The antimicrobial effects of heated oyster shells have been attributed to the generation of reactive oxygen species (ROS), such as hydroxide radicals (OH⋅) and hydrogen peroxide (H_2_O_2_), which contribute to protein denaturation and bacterial cell damage [13,14]. The presence of ROS, including HO_2_•, •O_2_, and H_2_O_2_, in aqueous solutions of shell-derived antimicrobial materials further supports the role of these species in antimicrobial mechanisms [15,16]. Among the oxygen-containing species, singlet oxygen (^1^O_2_) is known to be the most reactive [17]. It has been reported that singlet oxygen induces oxidative stress, leading to bacterial membrane protein damage [18]. According to [19], changes in cell membrane permeability are a critical factor in the antimicrobial effects of heated shells, and this effect varies depending on the bacterial cell wall structure. Their findings further provide evidence that singlet oxygen plays a major role in the antimicrobial mechanism of heated oyster shells.

The oxygen-containing species generated from heated oyster shells are thought to be formed through the well-known “lime cycle,” which describes the chemical transformations of calcium compounds [20]. Calcium exists in three interconvertible forms—CaCO_3_, CaO, and Ca(OH)_2_—depending on environmental conditions. Specifically, calcium carbonate (limestone) is thermally decomposed (lime burning) into calcium oxide (quicklime) at high temperatures. The resulting calcium oxide readily reacts with water (lime slaking) to form calcium hydroxide (hydrated lime). Subsequently, calcium hydroxide can undergo carbonation in the presence of CO_2_, reverting to calcium carbonate as a result. Therefore, the conversion of calcium carbonate in heated shells to calcium oxide is believed to enhance the oxidative antimicrobial properties of the heated oyster shells. However, despite these promising properties, there is still limited information on how these materials perform practically within real food systems and what physicochemical factors most strongly influence their antimicrobial efficacy. Particularly, the role of ion release, solubility, and surface characteristics in practical food matrices remains unclear.

If processed oyster shell powders, primarily intended as calcium-supplying food additives, simultaneously exhibit antimicrobial properties, this dual functionality could significantly enhance the overall value of oyster shell upcycling [21]. Therefore, in this study, we aimed to evaluate the antimicrobial activity of thermally processed oyster shell powder (TPOS) and citric acid-treated oyster shell powder (TPOSc) by investigating their physicochemical characteristics and practical applicability as food additives. The effects of citric acid treatment on neutralizing alkalinity, as well as the implications of physicochemical changes on antimicrobial efficacy, were specifically explored. Ultimately, this study seeks to establish the foundational knowledge necessary to transform oyster shell waste into effective, food-grade antimicrobial materials, contributing simultaneously to environmental sustainability and food safety.

## 2. Materials and Methods

### 2.1. Thermal Processed Oyster Shell Production

Except for coral-derived calcium supplements, all oyster shell-derived calcium materials used in this study were supplied by Vovi C&E (Yeongwol, Kangwon, Republic of Korea), a company specializing in the production of calcium-based materials derived from oyster shells using patented processing technologies (KR Patent No. 10-1405431). The oyster shells were harvested between 2015 and 2017 from the Yeongwol, Gangwon Province, Korea, washed using high-pressure hydrothermal treatment, and dried at 120 °C for over 24 h. The dried oyster shells were pulverized using a grinder and sieved through a standard 80-mesh sieve, which was followed by two thermal processing steps. The first thermal treatment was conducted at ≥750 °C in an air atmosphere for 5 h. Subsequently, the second thermal treatment was carried out in a nitrogen-oxygen mixed gas atmosphere at ≥600 °C for 6 h [22]. Through this process, TPOS was obtained. To enhance its functionality, 12% (*w*/*w*) citric acid was added to produce citric acid-treated TPOS (TPOSc). Fibrous calcium carbonate (FCC), supplied by Vovi C&E, was produced by oxidatively calcining CaCO_3_ to achieve a CaO conversion ratio of approximately 30–40%, which was followed by reductive calcination with magnesium. The final product exhibited a fibrous morphology with a length/diameter ratio of approximately 6:1, hence its designation as fibrous calcium carbonate. Additionally, as reference material, commercially available coral calcium supplement (CCP) was purchased from NOW Foods (Bloomingdale, IL, USA) for comparative analysis.

### 2.2. Analytical Methods

#### 2.2.1. Particle Size Analysis

The particle size distribution and dispersion of the samples were measured using dynamic light scattering (DLS) (Zetasizer Nano S90, Malvern, Worcestershire, UK). Each sample was analyzed in eight replicates using the same procedure, and the mean values were calculated for comparison. The polydispersity index (PDI) indicates the degree of particle size dispersion. A PDI value closer to 0 indicates homogeneity and a value closer to 1 indicates heterogeneity.

#### 2.2.2. Solubility Measurement

The solubility was determined based on the extent of dissolution in distilled water. Eight samples were prepared for each type, and the average values were calculated for comparative analysis. A 0.1 g sample was mixed with 50 mL of distilled water and left undisturbed for 10 min. The mixture was agitated at 8000 rpm using a homogenizer (HG-15A main unit equipped with HT1025 mixing head, Witeg Labortechnik GmbH, Wertheim, Germany) for 15 min and filtered using No. 42 Whatman filter paper (2.5 μm). The filtrate was transferred into pre-weighed vials (tared to a constant weight), and the weight of the vial with filtrate was recorded. The samples were subsequently dried at 100 °C for 12 h and re-weighed until a constant weight was achieved. In this study, a gravimetric method was employed to evaluate the relative solubility of the sample. After filtration, 10 mL of the filtrate was dried, and the weight of the remaining solid, including the container, was measured. The solubility was then calculated by subtracting the tare weight of the container from the weight after drying, then dividing by the initial weight of the filtrate [23]. Solubility was calculated using the following equation, Equation (1):
(1)$$Solubility\;\left(\frac{mg}{g}\right)=\frac{Weight\;of\;dried\;filtrate}{Weight\;of\;filtrate}$$

To assess the significance of solubility differences, calcium oxide (EP, 96%), calcium hydroxide (EP, 95%), and calcium carbonate (EP, 98%) were analyzed under the same conditions for comparison.

#### 2.2.3. pH Measurement of Calcium Samples

To evaluate the pH changes in response to sample addition, 0.5 g of each sample was mixed with 100 mL of distilled water (pH 4.5) and the pH was measured every 10 s for 10 min using a pH meter (Orion 3-Star Plus, Thermo Scientific, Waltham, MA, USA).

#### 2.2.4. SEM-EDS Analysis

The surface morphology and elemental composition of the samples before and after dissolution were analyzed using scanning electron microscopy (SEM) and energy-dispersive X-ray spectroscopy (EDS). Untreated samples and those immersed in 50 mL of distilled water for 24 h at 65 °C were used for analysis. The samples were mounted on carbon tape and coated with platinum using a sputter coater (108 Auto, Ted Pella Inc., Redding, CA, USA) before SEM observation (FIB LYRA3, TESCAN, Brno, South Moravian Region, Czech Republic).

#### 2.2.5. X-Ray Diffraction (XRD) Analysis

X-ray diffraction (XRD) analysis was performed using a LabX XRD-6100 (Shimadzu, Kyoto, Japan) under the following conditions: scanning rate of 2°/min and a 2θ scanning range of 10–80°. Samples included untreated powders and those immersed in 50 mL of distilled water for 24 h at 65 °C, which was followed by drying for 24 h.

### 2.3. Evaluation of Antibacterial Activity

Each sample was suspended in Luria-Bertani (LB) broth, Miller (Difco, Detroit, MI, USA), at final concentrations of 0.05, 0.1, 0.3, 0.5, and 0.7% (w/v) and stirred at 400 rpm. A pre-cultured *E. coli* suspension was inoculated into LB broth and incubated for 8 h. The pre-cultured bacteria were inoculated into the prepared sample suspensions and incubated, and incubation was conducted at 37 °C for 0, 1, 2, 4, 6, 8, 10, 12, and 24 h. At each time point, the samples were serially diluted and plated onto Plate Count Agar (PCA) (Difco, Detroit, MI, USA), which was followed by incubation at 37 °C for 24 h. LB broth without added samples served as the negative control.

### 2.4. Effects of Calcium Sample Application on Rice Cake

#### 2.4.1. Sample Preparation

To evaluate the effects of calcium treatment on rice cakes, various analyses including antimicrobial activity were conducted following sample preparation. Rice cake was prepared by mixing 200 g of rice with 160 mL of water and 0.1, 0.3, or 0.5 wt% of each calcium sample, based on the total weight of the rice and water mixture. The rice mixture was cooked in a rice cooker and subsequently extruded into rice cakes. The rice cakes were cut into 25 g portions, sealed in zipper bags, and stored at room temperature (25 ± 2 °C).

#### 2.4.2. Antifungal Activity Test

Fungal growth was monitored on the surface of the rice cakes over a 9-day period. The samples were stored at room temperature (25 °C) under ambient conditions throughout the observation period. Each sample was observed daily, and the day on which visible mold first appeared was recorded. Mold growth was assessed visually without magnification, and the antifungal effectiveness of each treatment was ranked based on the delay in the onset of visible mold formation.

#### 2.4.3. Color and pH Changes

##### Color Measurement of Rice Cake Samples 

Rice cake color was measured using a chromameter (CR-400, Konica Minolta Inc., Osaka, Japan). A white calibration plate (*L** = 86.40, a* = 0.3162, *b** = 0.3230) was used for calibration. Color changes were evaluated based on *b** values, where increased *b** indicated yellowing. The intensity of yellowing was qualitatively compared among samples at each calcium concentration level [24]. The yellowness index (YI) was calculated using the following equation, Equation (2), and monitored for three days:
(2)$$YI=\frac{142.86\times b}{L}$$

##### pH Measurement of Rice Cake Samples

The pH of rice cake samples was measured using a pH meter. For analysis, 5 g of each rice cake sample was homogenized with 45 mL of distilled water using a stomacher (BagMixer 400, Interscience Laboratory Inc., St. Nom, France) for 2 min. The homogenate was allowed to stand briefly, and the pH was measured directly from the supernatant.

#### 2.4.4. Antimicrobial Activity Test

The antimicrobial activity of calcium-treated rice cakes was evaluated by monitoring the natural microbial growth over time. The rice cake samples were stored at room temperature (25 °C) under ambient conditions throughout the test period to simulate typical storage environments. For each treatment, 5 g of rice cake was homogenized with 45 mL of sterile peptone water using a stomacher for 2 min. The homogenate was serially diluted with sterile peptone water, and 1 mL of the appropriate dilution was plated onto PCA using the pour plate method. The plates were incubated at 37 °C for 24 h, and the total viable count was expressed as log CFU/mL. Microbial growth was assessed daily for three days to evaluate the antimicrobial effect of each calcium-treated rice cake sample.

#### 2.4.5. Texture Analysis of Rice Cake

The texture of the rice cakes was analyzed using a texture analyzer (TA-XT, MHK Corp., Anyang, Gyeonggi, Republic of Korea) equipped with a 5 kg load cell. Samples were prepared in cylindrical shapes with a diameter of 1.2 cm and a height of 1 cm. Hardness and chewiness were evaluated using the texture profile analysis (TPA) method. The test was conducted using a special test mode with the following conditions: pre-test speed of 1.0 mm/s, test speed of 1.0 mm/s, and post-test speed of 2.0 mm/s. The strain was set at 85%, and the trigger force was set to 5 g. Measurements were taken in triplicate, and the results were analyzed to determine hardness and chewiness. This method was adapted based on previous studies with similar starch-based food matrices [25,26].

## 3. Results and Discussion

### 3.1. Analytical Results

#### 3.1.1. Powder Characteristics

To understand the fundamental properties of the calcium samples, their physical characteristics—such as their appearance, tactile texture, particle size, and PDI—were analyzed. All calcium samples were odorless powders. TPOS and TPOSc exhibited a grayish color, whereas FCC and CCP appeared white. In terms of tactile sensation, the smoothness of the powders was ranked as follows: CCP < TPOSc < TPOS < FCC. Among the oyster shell-derived calcium samples, TPOSc was perceived to have a more uniform texture compared to TPOS. FCC, unlike the other powdered samples, had a crumbly texture and appeared as loosely aggregated granules. To further analyze the observed powder characteristics, particle size and PDI were measured using DLS. All samples exhibited relatively narrow particle size distributions. The smallest particle size was observed in FCC, which had an average diameter of approximately 476 nm and the lowest standard deviation (48.910). The oyster shell-derived calcium products, TPOS and TPOSc, had an average particle size of approximately 1000 nm. However, TPOSc exhibited a slightly larger particle size and PDI compared to TPOS. The coral-derived calcium product, CCP, had an average particle size nearly twice that of the oyster shell-derived products and displayed a higher degree of PDI. These findings corresponded well with the sensory evaluation results described earlier in this section (Table 1).

#### 3.1.2. Solubility Analysis

The solubility of a substance is influenced not only by its chemical composition but also by its particle size, particle size distribution, and surface area. Even substances that are generally considered insoluble in water can exist in a dispersed state if their particle size is sufficiently small, forming a stable colloidal suspension that behaves similarly to a true solution [27].

The solubility of three calcium compounds involved in the lime cycle varies significantly: CaCO_3_ has a solubility of 0.013 g/L at 25 °C, while CaO immediately reacts with water to form Ca(OH)_2_, which has a solubility of 1.73 g/L at 20 °C. Thus, calcium carbonate is largely insoluble in water, whereas calcium oxide and calcium hydroxide are relatively more soluble [28,29].

As observed in the particle size analysis, the experimental samples, with particle sizes ranging from 300 to 1000 nm, were likely to exist in a suspension state [30]. In contrast, larger particles tend to settle over time, which potentially reduces their bioavailability [31]. To quantify and compare the proportion of calcium compounds that dissolves or remains in a stable suspension, solubility was measured. For validation, commercially available CaO (96% EP), Ca(OH)_2_ (95% EP), and CaCO_3_ (98% EP) were analyzed using the same method. As shown in Figure 1, the measured solubilities of these reference compounds closely matched their known solubilities, confirming the validity of the experimental method. Figure 1 shows that TPOS and TPOSc exhibited a solubility of approximately 0.5 and 0.4 mg/g, respectively, which are significantly higher than those of pure calcium carbonate but approximately one-third that of calcium hydroxide. In contrast, FCC, a calcium oxide-based product, had the highest solubility among the experimental samples at approximately 0.7 mg/g. This was 1.5 times higher than that of the oyster shell-derived calcium compounds but only 50% of the solubility of pure calcium oxide. This high solubility is likely attributable to the partial conversion of FCC to reactive calcium oxide during calcination, along with its fibrous morphology that increases its surface area and facilitates more efficient dissolution in aqueous environments. Meanwhile, CCP exhibited the lowest solubility at 0.05 mg/mL, which is higher than that of pure calcium carbonate but significantly lower than those of the other experimental samples.

Taken together, these results suggest that the experimental samples are not composed solely of pure calcium carbonate, calcium oxide, or calcium hydroxide. Additionally, the findings align with the particle size analysis, indicating that smaller calcium compound particles tend to have higher solubility. Notably, TPOS, TPOSc, and FCC, which exhibited higher solubility than the commercially available coral calcium product (CCP), may be more suitable for use as calcium supplements due to their potentially higher bioavailability.

#### 3.1.3. pH Changes in Aqueous Solutions

When calcium-based samples dissolve or remain in suspension, one of their most significant chemical characteristics is their effect on the pH of the solution. These pH changes reflect the chemical properties of the substances and provide fundamental insights into their reactivity with other compounds [32].

As shown in Figure 2, TPOS and FCC reached stabilized pH values of approximately 12.3 and 12.5, respectively, within 60 s. CCP attained pH 9.8 after 60 s, with no further significant changes being observed thereafter. In contrast, TPOSc exhibited a relatively gradual increase in pH, reaching around pH 10 after approximately 6 min.

While TPOS and FCC exhibited similar strong levels of alkalinity, CCP and TPOSc showed final pH values that were mildly alkaline, which clearly distinguished them from the former group. These findings indicate that the samples have distinctly different chemical and functional characteristics. This implies that pH changes result not only from dissolution but also from ionization and hydroxide ion (OH^−^) release, which are influenced by both the chemical nature and concentration of the calcium compounds.

Based on these results, we conducted a follow-up experiment using 50% acetic acid (initial pH 1.4) instead of distilled water. The pH changes were monitored over time after the addition of the calcium-based samples to observe how the pH responded in the acidic environment (Figure 3). All four samples exhibited a time-dependent increase in pH, which indicated that the calcium materials dissolved and neutralized H^+^ ions in the acidic solution. However, unlike the patterns observed in DI water, the pH elevation trends in acetic acid differed across the samples. These differences suggest that the extent of ionization varies among the samples, and that alkalinity is more closely related to the ion-releasing capacity than to dissolution alone.

Notably, the citric acid-coated TPOSc exhibited pH buffering and stabilization behavior, indicating that further investigation is required to elucidate its underlying mechanisms.

#### 3.1.4. Characteristics After Dissolution

Based on the previously observed and analyzed results, it was deemed necessary to compare and analyze the particle states of the calcium compound samples both in their powdered form and after dissolution. The surface characteristics of materials have been reported to be a critical factor influencing their antimicrobial activity [10]. Therefore, SEM was used to observe changes in surface morphology, while EDS was employed for elemental analysis. Additionally, by comparing the samples in their original powdered state with those subjected to aqueous treatment and subsequent drying, this study aimed to elucidate the physicochemical and structural characteristics underlying the observed solubility increase and antimicrobial properties.

SEM analysis of TPOS (Figure 4), a thermally processed oyster shell powder, revealed a smooth calcite-like structure. However, the particle size and shape were irregular, with a heterogeneous distribution of particles of varying sizes. EDS analysis of the powdered TPOS sample indicated that, apart from magnesium (Mg), the structure was predominantly composed of calcium compounds.

After exposure to an aqueous environment and subsequent drying, SEM and EDS analyses were performed on TPOS. While no significant changes were detected in its elemental composition via EDS, SEM imaging showed the formation of a porous structure on the particle surfaces, with similarly sized particles clustering together. These observations suggest that TPOS consists not only of CaCO_3_ but also of CaO or Ca(OH)_2_. Additionally, the particles appeared to become more uniform in size and slightly more rounded, which implied that surface reactions had occurred, resulting in minor morphological changes. It remains unclear whether these changes occurred during suspension in the aqueous medium or upon drying after dissolution. However, the presence of surface reactions upon contact with water is evident. This observation is further supported by changes in the Ca/C/O mass ratio, which shifted from approximately 1:3:7 to 1:4:6, suggesting the leaching of calcium ions during aqueous treatment.

TPOSc (Figure 5), a citric acid-treated derivative of TPOS, exhibited a unique surface morphology. SEM imaging of the powdered TPOSc revealed small particles adhering to the surface, resembling a coated structure. This is likely due to the encapsulation effect of citric acid, which is commonly used as a stabilizing agent for metallic and non-metallic particles [33]. The citric acid coating prevented migration, adsorption, or agglomeration of calcium carbonate or calcium oxide particles, and thereby stabilized the material.

This hypothesis was further validated by comparing SEM images of TPOSc before and after aqueous exposure. In the SEM images of TPOSc subjected to aqueous treatment and subsequent drying, the previously dispersed small particles were observed to aggregate, forming clusters. Additionally, a smoother surface was noted, suggesting that the citric acid coating was removed after aqueous treatment and drying, and exposing the calcite structure of calcium carbonate. The presence of mixed calcite and aragonite structures was inferred from these observations.

Such findings have significant implications for the functional properties of calcium supplements or additives. The chemical functionality of calcium particles, including their alkalinity and reactivity, is influenced by the exposure of surface ions such as Ca^2+^, HCaO^−^, and OH^−^. If citric acid enhances particle stability or modulates pH changes, then the physicochemical properties of TPOSc may differ from those of conventional calcium compounds. This hypothesis warrants further investigation using zeta potential measurements and Fourier transform infrared (FT-IR) spectroscopy.

Notably, silicon (Si), which was not detected in TPOS, was identified in TPOSc in trace amounts, likely due to adsorption during processing. However, the concentration was negligible, which suggests minimal impact on the material’s functionality.

FCC (Figure 6), a calcium carbonate nanoparticle, was subjected to thermal calcination that converted approximately 40% of its content to calcium oxide, as a result of which it exhibited a clean surface with dispersed small particles in its powdered form. SEM imaging typically does not differentiate between calcium carbonate and calcium oxide. However, after aqueous treatment and drying, FCC displayed a unique net-like structure. Given FCC’s high solubility and rapid pH changes, this surface transformation may be closely related to its functional properties. Further research is required to determine whether these structural changes are due to crystallographic modifications or molecular rearrangement. In contrast, the coral-derived calcium product (CCP) (Figure 7) showed no significant difference in SEM analysis between the powdered form and the aqueous-treated state. This observation aligns with the previously reported results for solubility and antimicrobial activity, supporting the notion that CCP exhibits relatively low reactivity in aqueous environments. FCC, TPOS, and TPOSc underwent distinct structural changes upon aqueous exposure—such as net-like or porous surface formation—which were accompanied by calcium ion release. These ions likely interact electrostatically with fungal membranes, which leads to structural disruption and antimicrobial effects. Additionally, calcium may disturb the ionic homeostasis inside the cells [12,34].

#### 3.1.5. XRD Analysis

To determine whether the observed results were influenced by changes in calcium compound crystal structures or states, X-ray diffraction (XRD) analysis was performed. Previous studies on the thermal processing of oyster shells have shown characteristic XRD pattern shifts corresponding to the formation of calcium oxide at high temperatures (Figure 8). Calcium carbonate primarily exists in three crystal structures—calcite, aragonite, and vaterite—along with three additional structures (monohydrocalcite, ikaite, and amorphous calcium carbonate), which results in six known polymorphic forms. The three main crystal structures exhibit distinct XRD patterns. By analyzing the XRD patterns of each sample, differences in crystalline structures were examined. Additionally, samples treated with aqueous solutions and subsequently dried were analyzed to assess structural changes induced by environmental factors, particularly water exposure. Phase identification was carried out by comparing the obtained XRD patterns with reference data from the Powder Diffraction File (PDF) database maintained by the International Centre for Diffraction Data (ICDD).

In the case of TPOS (Figure 9A), the XRD pattern of the untreated powder sample showed characteristic peaks assigned to calcite-type calcium carbonate, calcium hydroxide, and CaO (PDF 04-004-8985). After aqueous treatment and drying (Figure 9B), the CaO peak remained, but the reference pattern shifted to PDF 04-016-3215, which indicated a change in the crystalline form. This suggests that, while CaO was not removed, partial dissolution and reprecipitation or surface rearrangement during the drying process may have led to structural transformation [35].

In the case of TPOSc (Figure 9C,D), the XRD pattern of the untreated sample showed characteristic peaks assigned to CaCO_3_ and Ca(OH)_2_. However, after aqueous treatment and drying, only the calcite phase was detected, and the calcium hydroxide peak became undetectable. This indicates that the hydroxide component may have dissolved in the solution or undergone structural transformation beyond the detection limit of the analysis.

Notably, while CaO was clearly detected in TPOS (Figure 9A,B), CaO peaks were negligible or undetectable in both untreated and treated TPOSc samples (Figure 9C,D), a closer examination suggests that a small proportion of CaO may have initially been present in the untreated state (Figure 9C), which then converted primarily into calcium hydroxide upon aqueous treatment (Figure 9D). This absence suggests that the citric acid treatment may have inhibited CaO formation during thermal processing, potentially by modifying the local pH environment or promoting alternative calcium salt formation. In contrast, TPOSc showed a distinct transformation pattern. XRD analysis revealed that, while untreated TPOSc contained a relatively high proportion of calcium oxide (Figure 9C), this was converted primarily into calcium hydroxide following aqueous treatment (Figure 9D), with the calcium oxide becoming undetectable. This phase transformation is likely influenced by citric acid, which may alter surface-level reaction kinetics, prevent complete conversion to CaO during thermal processing, or delay re-carbonation after aqueous exposure. In the FCC (Figure 9E,F), the disappearance of CaO peaks after aqueous treatment was accompanied by a marked increase in Ca(OH)_2_ peak intensity, which suggests that the CaO underwent hydration to form calcium hydroxide. This transformation is consistent with the known slaking behavior of lime in the presence of water, indicating structural reformation rather than complete dissolution. The presence of CaO in FCC is attributable to its production process, in which approximately 40% of the original calcium carbonate is thermally converted to calcium oxide, as described in Section 2.1. For CCP (Figure 9G,H), only calcium carbonate was detected, with no significant changes being observed either before or after the aqueous treatment. These results suggest that surface CaO is present in TPOS, TPOSc, and FCC which was converted to Ca(OH)_2_ upon water exposure. These structural phase transitions are consistent with the solubility and pH behaviors observed earlier.

### 3.2. Antibacterial Activity

Calcium carbonate (CaCO_3_) and CaO are known to exhibit antimicrobial activity due to their alkaline-induced effects, which disrupt microbial cell membranes and active transport mechanisms [36]. To determine whether the experimental calcium samples possessed antibacterial activity, a series of antimicrobial tests were conducted. *E. coli* was selected as the target microorganism, and liquid media assay was used to evaluate the antimicrobial activity of the oyster shell-derived calcium carbonate samples.

As shown in Figure 10A, in the case of bacterial cultures incubated with TPOS suspensions, no bacterial growth was observed at concentrations of 0.5% and 0.7%. At 0.3%, the bacterial growth remained stable over the first 12 h, which suggests growth suppression or a limited bactericidal effect. A subsequent decline by 24 h indicates delayed bacterial death. At the lower concentration of 0.1%, no inhibitory effect on bacterial growth was observed. For cultures incubated with TPOSc suspensions (Figure 10B), the bacterial growth was maintained at 0.5%, while no growth was observed at 0.7%, which indicated a lower antibacterial activity compared to TPOS. In contrast, the FCC suspensions (Figure 10C) showed the highest antibacterial activity, with complete bacterial death being observed at concentrations of 0.3% or higher.

In the case of CCP (Figure 10D), no inhibitory effect on bacterial growth was observed even at the highest tested concentration of 0.7%, which indicated the weakest antibacterial performance among the tested samples. These results suggest that TPOS and FCC, which exhibited relatively higher pH values upon dissolution, also demonstrated the strongest antibacterial activities. However, TPOSc, which maintained a near-neutral pH similar to that of CCP, still exhibited notable antibacterial activity, unlike CCP.

These findings suggest that factors beyond the increase in pH due to calcium dissolution contribute to the observed antimicrobial activity. As previously discussed, the increased release of calcium ions (Ca^2+^) may interfere with bacterial metabolic processes, resulting in antimicrobial effects through a combination of mechanisms [37].

Additionally, particle size appears to play an important role in the antimicrobial efficacy of calcium carbonate. Calcium carbonate at the microscale exhibits no antimicrobial effect, whereas its nanoscale form shows significant antibacterial activity. This is attributed to the increased surface area of smaller particles, which enhances the contact with bacterial membranes, allowing for attachment or penetration and ultimately leading to physical damage and cell death [38]

Therefore, calcium powders derived from oyster shells exhibit effective antimicrobial properties and hold potential as sustainable and eco-friendly antimicrobial agents. These findings suggest that such materials may serve as viable alternatives to conventional antimicrobial compounds.

### 3.3. Effects of Sample Application on Rice Cake

#### 3.3.1. Color Changes and Antifungal Properties

Rice cake was selected as the test food product due to its relatively simple manufacturing process and high susceptibility to microbial spoilage. As it is prepared using glutinous rice, which provides a carbohydrate-rich environment, it is particularly prone to mold growth under ambient storage conditions [39,40]. Due to its high susceptibility to microbial spoilage and simple composition, rice cake was selected as a suitable model food matrix for evaluating the effects of calcium-based samples on pH, discoloration, and fungal contamination under ambient storage conditions.

As shown in Figure 11A, when each sample was added to the rice cake at a concentration of 0.1 wt%, the pH ranged between 6 and 7. According to Figure 11B, all samples maintained a white color, and no noticeable color change was observed. In terms of antifungal properties, no visible mold growth was observed on day 1 for any of the samples. However, by day 2, mold growth was detected in all samples. Therefore, at a concentration of 0.1 wt%, there were no significant differences among the samples in terms of either their color change or antifungal activity.

At a sample concentration of 0.3 wt%, noticeable differences were observed in the pH, color, and antifungal properties among the rice cake samples. The control and CCP-added rice cakes maintained pH levels that were similar to those observed at the lower concentration. In contrast, samples with TPOS and FCC exhibited elevated pH values of approximately pH 9, while the TPOSc sample showed a moderate increase to around pH 8. These pH changes were also reflected in the color of the rice cakes. Samples with relatively high pH values (TPOS and FCC) displayed a yellowish appearance, whereas the TPOSc and CCP samples retained their original white color. Regarding antifungal properties, mold growth was observed on day 2 in the control and CCP-added rice cakes. In comparison, the TPOS and TPOSc samples showed mold growth on day 3, and the FCC-added samples exhibited visible mold growth only after 7 days. Although these findings were based on visual observation rather than quantitative microbial analysis, a qualitative ranking of antifungal effectiveness can be suggested as follows: FCC > TPOS = TPOSc > CCP = control. This ranking reflects the relative delay in external mold growth observed during storage.

At a concentration of 0.5 wt%, the control and CCP-added rice cakes maintained similar pH levels with no significant changes. In contrast, the samples containing TPOS, TPOSc, and FCC showed elevated pH values, with the TPOSc-treated rice cake exhibiting a pH greater than 9. In terms of color, the control and CCP samples retained their original white appearance, while rice cakes containing TPOS, TPOSc, and FCC turned fully yellow. The intensity of the yellow coloration was more pronounced than that observed at the 0.3 wt% concentration. Regarding antifungal activity, mold growth appeared on day 3 in both the control and CCP-treated samples. For the TPOS and TPOSc groups, mold growth was first observed on day 6, while the FCC-supplemented rice cakes showed visible mold development only after 9 days. Accordingly, the qualitative ranking of antifungal effectiveness remained consistent with the previous results.

As the pH of the rice cakes increased with the addition of calcium-based samples, a consistent trend was observed in the color change, with higher pH values leading to more intense yellow coloration. This color shift was particularly evident in samples such as the FCC and TPOS. Moreover, a delay in mold growth was associated with these higher-pH and more strongly yellow-colored samples [41]. Among all samples, FCC demonstrated the most pronounced antifungal effect, followed by TPOS and TPOSc. These results indicate that, even when incorporated into a food matrix such as rice cake, calcium-based samples can retain their antifungal properties. The consistent trend across both the pH, color, and mold growth suggests that these parameters are interrelated, and that the alkaline nature of the samples may contribute to their antifungal efficacy. Considering the dense and solid nature of rice cake, which limits particle mobility and ion diffusion compared to aqueous systems, it is possible that the observed antifungal effects were primarily driven by the alkaline environment rather than by direct interactions between calcium ions or particles and microbial cells [42]. This may explain why pH increase showed a stronger correlation with mold inhibition in this food matrix.

#### 3.3.2. Antibacterial Properties in Rice Cake Matrix

As shown in Figure 12A, at a 0.1 wt% concentration, the initial bacterial count was approximately 1.5 log for the control and 2 log for the TPOSc, FCC, and CCP, while the TPOS exhibited a slightly higher count of 2.7 log. The bacterial growth rate was highest in the control from day 0 to day 1, but the increase was comparable among all samples. Even after two to three days, the bacterial counts of the samples remained within a 1 log difference from each other, suggesting only moderate effects on bacterial growth. These findings suggest that, at 0.1 wt%, the samples did not significantly affect the bacterial growth.

In Figure 12B, at a 0.3 wt% concentration, the initial bacterial count was approximately 1 log for the control and CCP, while it was below the detection limit for the TPOS, TPOSc, and FCC. Unlike the 0.1 wt% condition, bacterial growth was suppressed in the FCC-added rice cake, which exhibited only a 1 log increase over three days.

At 0.5 wt% (Figure 12C), the initial bacterial count across all samples was approximately 2 log CFU. However, in the TPOS- and FCC-treated rice cakes, the bacterial count remained unchanged over three days, while the CCP group exhibited a higher bacterial count than the control. In the case of TPOSc, the bacterial levels were lower than those of the control until day 2, but reached a similar level by day 3. Consistent with the trends observed in the pH and mold growth, these results indicate that rice cakes treated with TPOS and FCC maintained lower bacterial levels over an extended period, which is consistent with the delayed mold growth observed during the sensory evaluation. Similarly, TPOSc demonstrated antimicrobial activity, albeit to a lesser extent than TPOS and FCC, which reflects its behavior in the earlier aqueous suspension tests. In contrast, CCP showed an equal or even lesser antibacterial effect than the control.

Taken together, these results demonstrate that both TPOS and TPOSc retain effective antibacterial activity in food applications, even under solid matrix conditions. While their performance was somewhat lower than that of FCC, their dual functionality and upcycled origin highlight their potential as sustainable antimicrobial agents in functional food systems.

#### 3.3.3. Hardness and Chewiness Analysis

In the context of developing functional foods, it is essential to evaluate whether the addition of bioactive compounds, such as calcium-based antimicrobial agents, alters the fundamental texture properties of the product. Since texture significantly influences consumer acceptance, excessive changes in hardness or chewiness may lead to undesirable sensory perceptions and reduced product appeal. Therefore, this experiment aimed to assess the impact of calcium sample incorporation on the hardness and chewiness of rice cakes across different concentrations.

As shown in Figure 13A,B, at a 0.1 wt% calcium addition, rice cakes supplemented with TPOS, TPOSc, and CCP exhibited statistically significant differences in hardness compared to the control. However, the actual differences in values were relatively small, suggesting a limited impact on the perceived texture. In contrast, FCC showed a significantly higher hardness, which indicated a more pronounced textural effect at this concentration. As shown in Figure 13B, there were no statistically significant differences in chewiness among any of the sample groups at this concentration. At a concentration of 0.3 wt% (Figure 13C,D), all calcium-treated rice cakes exhibited lower hardness and chewiness values compared to the control. Significant intergroup differences were detected among all treated samples. Similarly, at 0.5 wt%, rice cakes containing TPOS, TPOSc, and FCC showed significantly reduced hardness and chewiness values, while those containing CCP did not show a significant difference from the control. This decrease in hardness may be attributed to smaller calcium particles or reactive species interacting with and partially replacing starch molecules in the rice cake matrix.

At 0.3 wt%, the chewiness was maintained within an acceptable range, which suggests a balance between texture and functionality. In particular, TPOS and TPOSc showed a favorable balance between antimicrobial functionality and texture preservation, which suggests their potential as promising candidates for functional food applications. These results indicate that calcium samples derived from oyster shells, when used at appropriate concentrations (≤0.3 wt%), can confer antimicrobial effects while preserving the desirable texture of rice cakes. While the antimicrobial properties of the calcium-treated rice cakes were the primary focus, the potential nutritional contribution of calcium was briefly considered. Based on the average solubility of TPOS (0.5 mg/g), and the applied concentration (0.3 wt%), a standard 100 g rice cake would contain approximately 150 mg of calcium-based material. Given the low solubility, the estimated available calcium would be around 75 mg per serving. Considering the recommended daily calcium intake for adults is about 1000 mg/day, this represents roughly 7.5% of the requirement. Therefore, although not sufficient as a standalone supplement, the inclusion of these materials may offer added value as a source of functional calcium, in addition to their antimicrobial properties. This highlights the potential of TPOS as a viable alternative material for the development of functional foods that not only support microbial control but also offer secondary nutritional benefits, such as supplemental calcium, without compromising consumer acceptability or excessively altering the product quality.

## 4. Conclusions

The growing aquaculture industry produces large volumes of oyster shell waste, which poses an increasing environmental challenge due to limited recycling and inadequate disposal methods. This study aimed to develop and evaluate antimicrobial agents derived from oyster shell-based calcium powders—namely TPOS, TPOSc, FCC, and CCP—by investigating their physicochemical characteristics and effectiveness in both controlled environments and real food matrices, such as starch-based rice cakes. The antimicrobial activity tests showed that TPOSc had meaningful antimicrobial effects at concentrations above 0.5%, which indicated that ion release and physical characteristics, in addition to alkalinity, also contribute to antimicrobial performance. To evaluate their applicability in actual food systems, the calcium-based materials were incorporated into starch-based rice cakes. FCC and TPOS showed stronger antibacterial and antifungal activities than TPOSc at concentrations above 0.3 wt%, which confirmed that their antimicrobial functionality was retained within real food matrices. These findings introduce a naturally derived antimicrobial agent suitable for food applications and provide a more nuanced understanding of how factors such as ion release, solubility, and surface characteristics, in addition to alkalinity, contribute to antimicrobial efficacy. Therefore, while this study demonstrated the clear antimicrobial potential of oyster shell-derived materials in a rice cake model system, the broader utility of these materials remains to be explored. Expanding this approach to other food systems could further validate the versatility and sustainability of these materials. Ultimately, this research lays the groundwork for transforming marine biowaste into value-added, functional food-preserving agents with practical industrial relevance.

## Figures and Tables

**Figure 1 foods-14-02579-f001:**
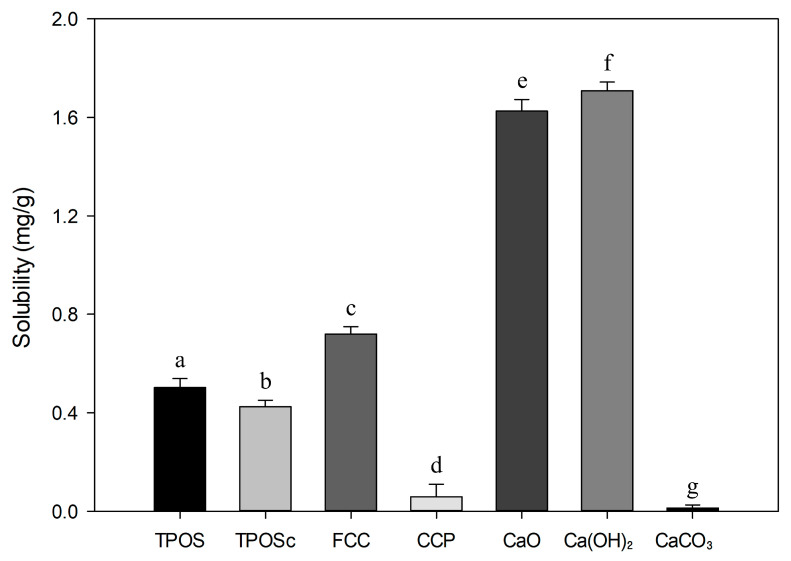
Solubility of samples and calcium compounds: Different letters above the bars indicate significant differences between groups according to Duncan’s test (*p* < 0.05).

**Figure 2 foods-14-02579-f002:**
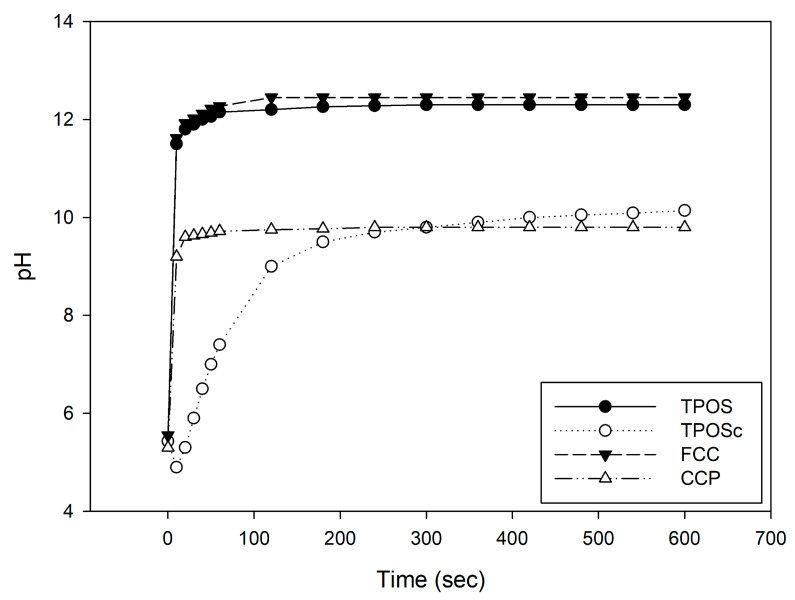
pH changes in calcium-based sample suspensions in DI water over time.

**Figure 3 foods-14-02579-f003:**
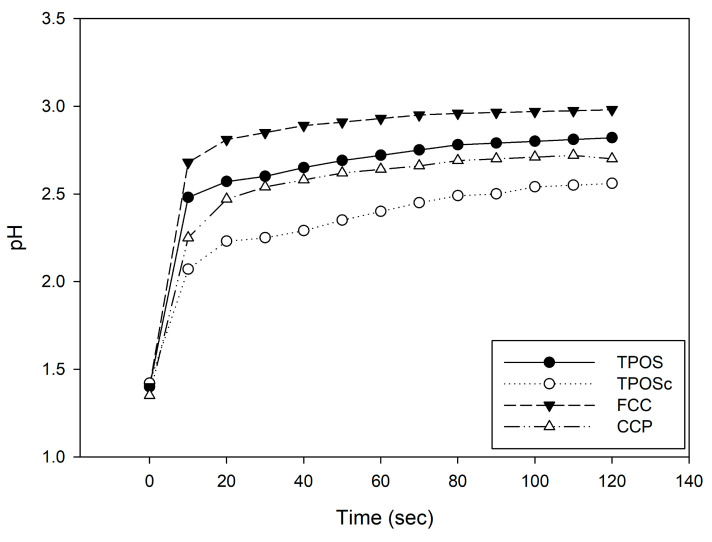
pH changes in calcium-based sample suspensions in 50% acetic acid solution (initial pH 1.4) over time.

**Figure 4 foods-14-02579-f004:**
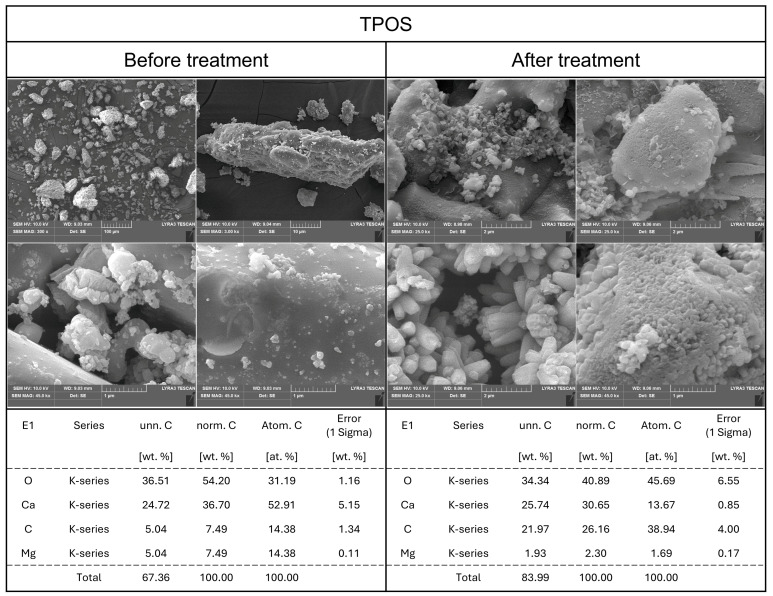
SEM-EDS analysis of TPOS before and after dissolving in DI water.

**Figure 5 foods-14-02579-f005:**
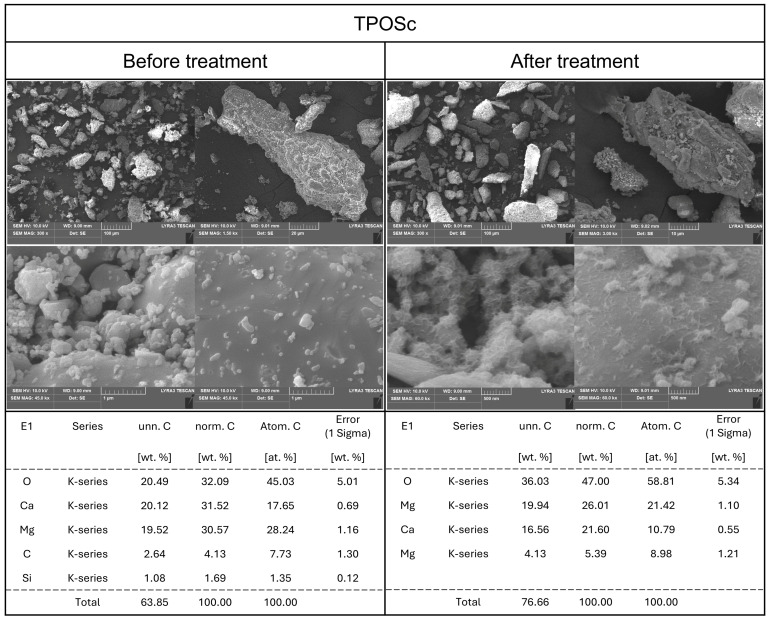
SEM-EDS analysis of TPOSc before and after dissolving in DI water.

**Figure 6 foods-14-02579-f006:**
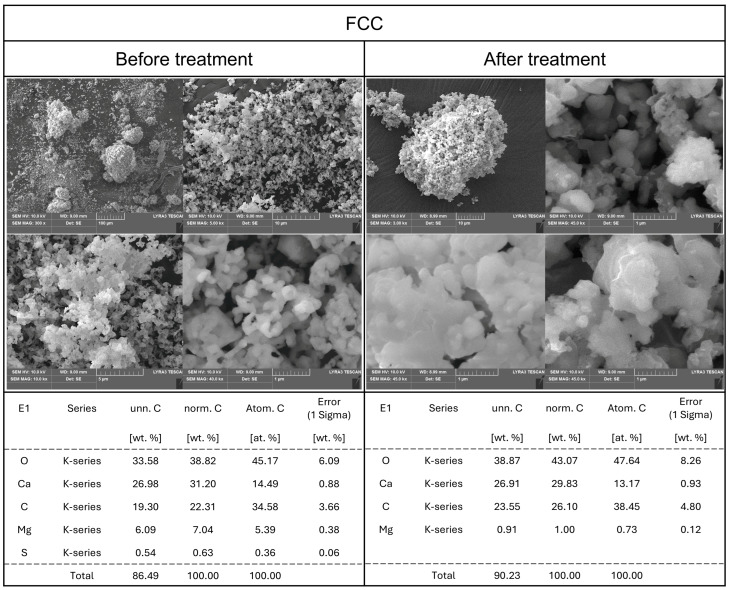
SEM-EDS analysis of FCC before and after dissolving in DI water.

**Figure 7 foods-14-02579-f007:**
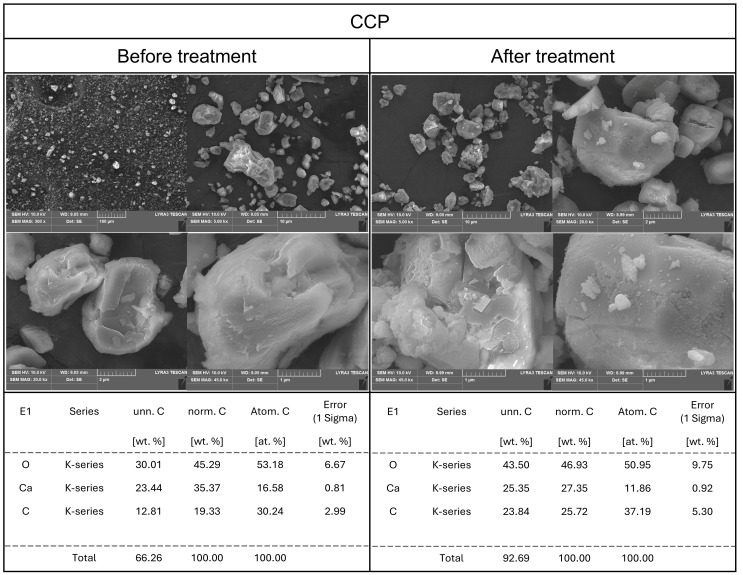
SEM-EDS analysis of CCP before and after dissolving in DI water.

**Figure 8 foods-14-02579-f008:**
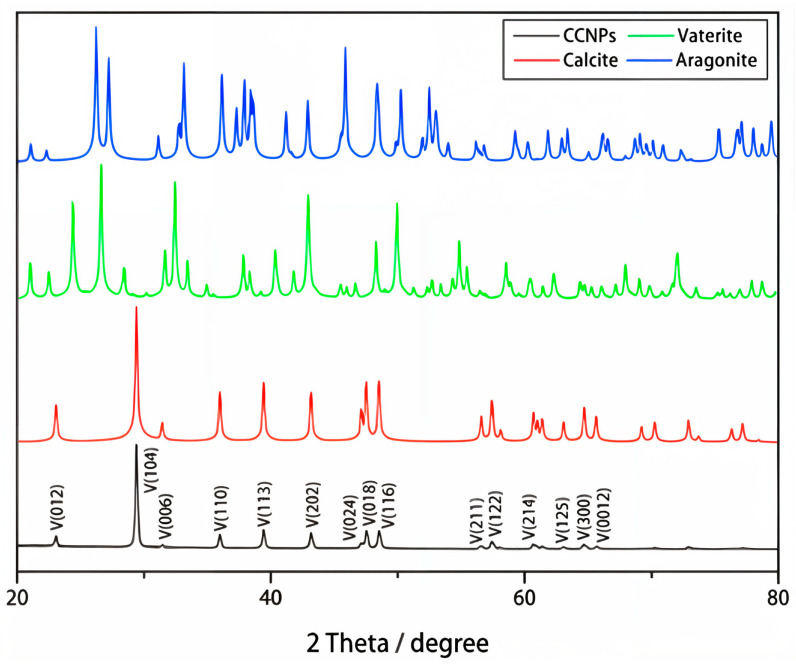
XRD analysis of polymorphic calcium carbonate: CCNPs, calcium carbonate nanoparticles.

**Figure 9 foods-14-02579-f009:**
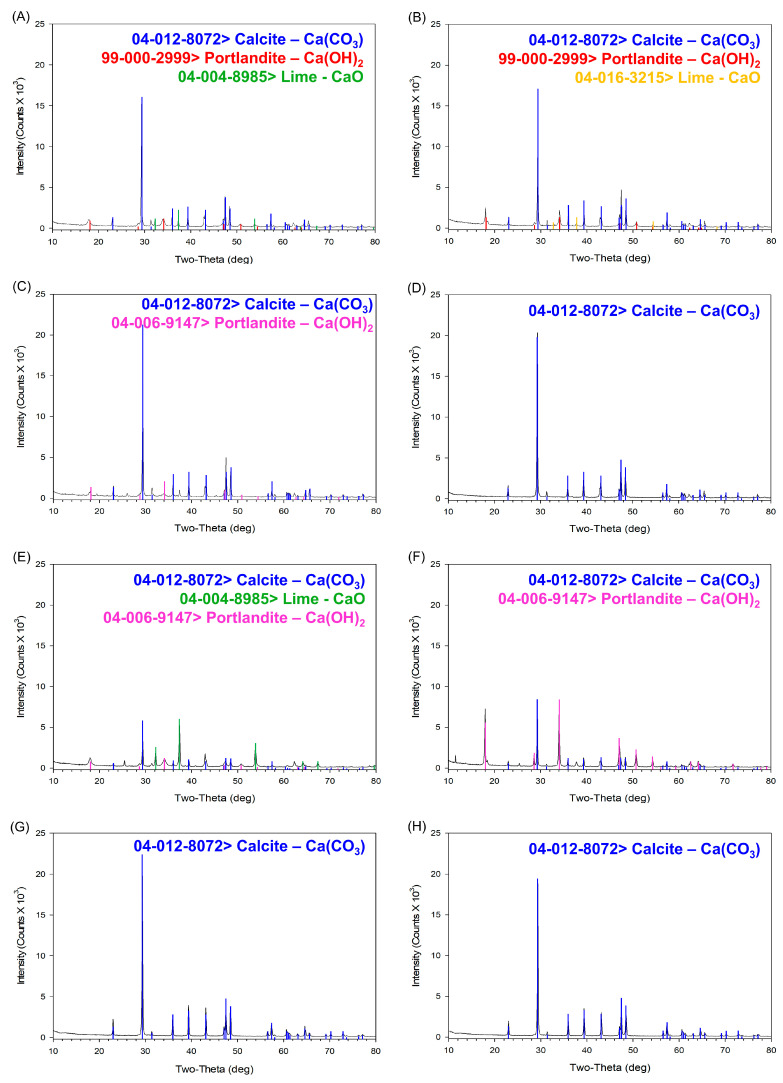
XRD patterns of calcium-based samples before and after dissolving in DI water: (**A**,**B**) TPOS, (**C**,**D**) TPOSc, (**E**,**F**) FCC, (**G**,**H**) CCP. Each pair of subfigures shows the XRD spectra of the respective sample before (left) and after (right) dispersion in distilled water, highlighting potential phase changes or crystallinity differences.

**Figure 10 foods-14-02579-f010:**
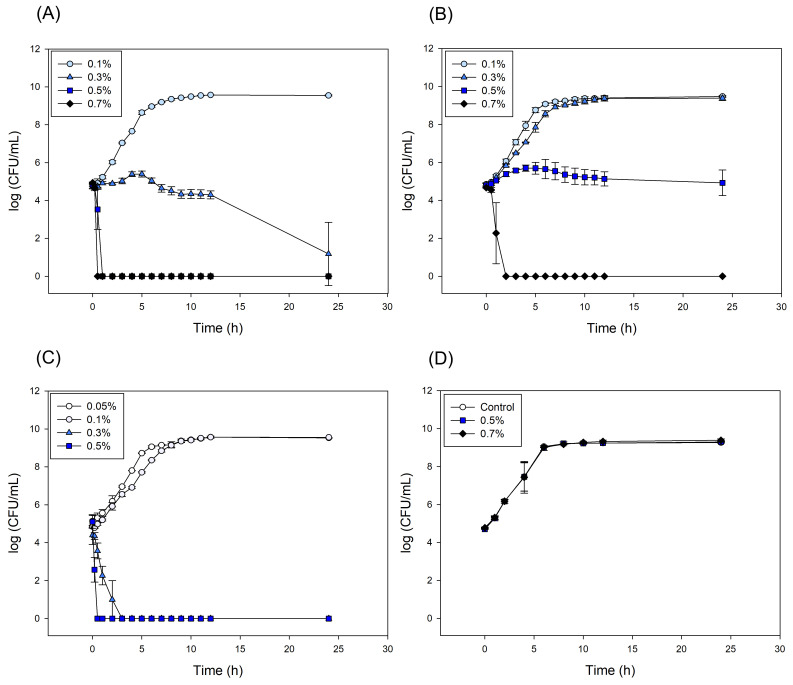
Antimicrobial effects of calcium-based samples against *Escherichia coli*: (**A**) TPOS, (**B**) TPOSc, (**C**) FCC, (**D**) control and CCP.

**Figure 11 foods-14-02579-f011:**
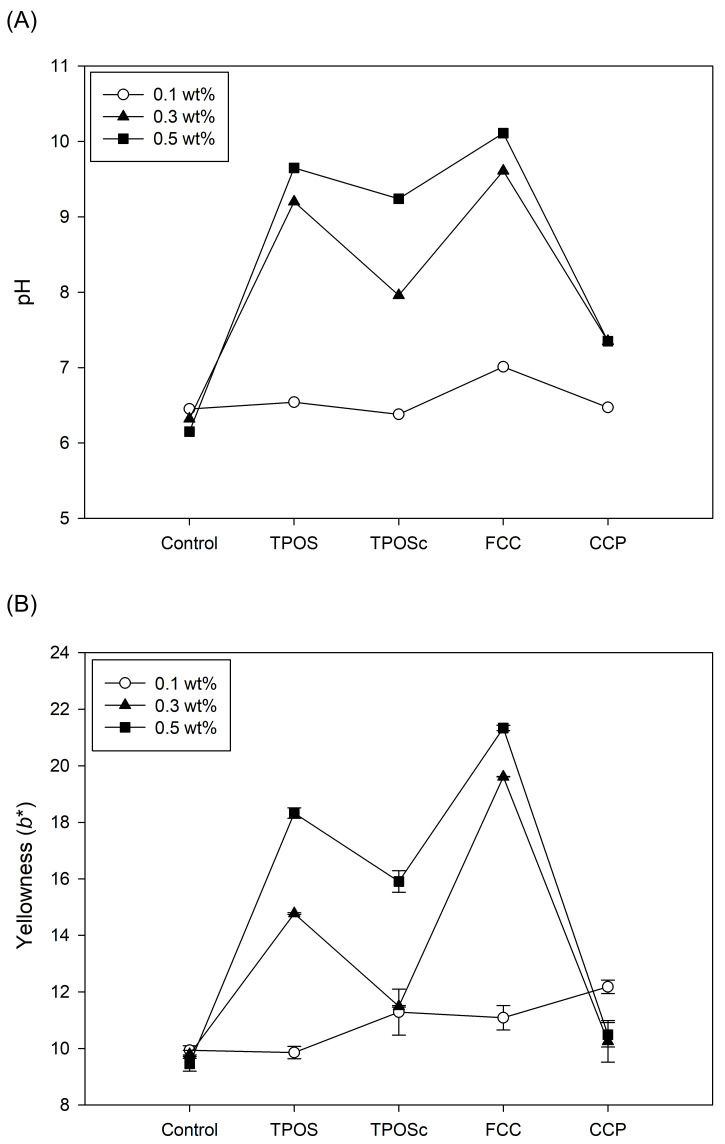
Effects of calcium-based samples addition on the (**A**) pH and (**B**) yellowness (*b**) values of rice cakes at different concentrations (0.1, 0.3, and 0.5 wt%).

**Figure 12 foods-14-02579-f012:**
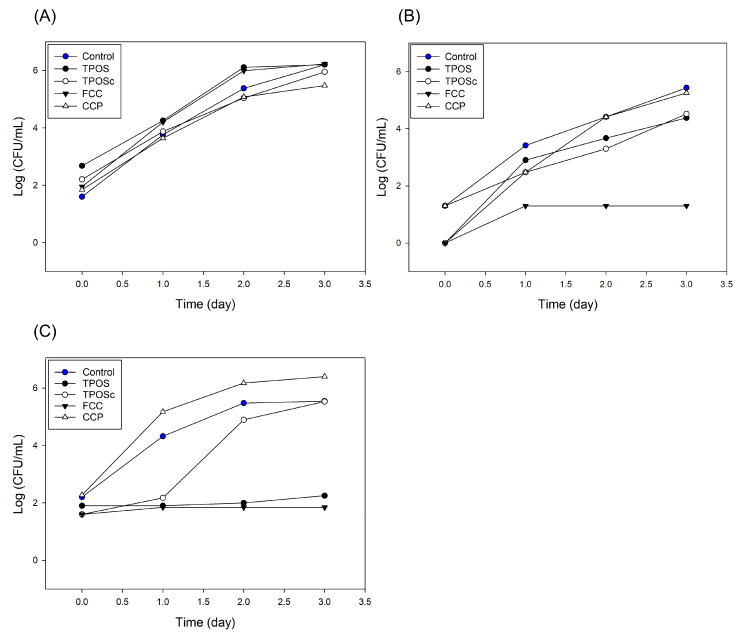
Changes in natural microbial levels in rice cakes supplemented with calcium-based samples over three days at concentrations of (**A**) 0.1 wt%, (**B**) 0.3 wt%, and (**C**) 0.5 wt%.

**Figure 13 foods-14-02579-f013:**
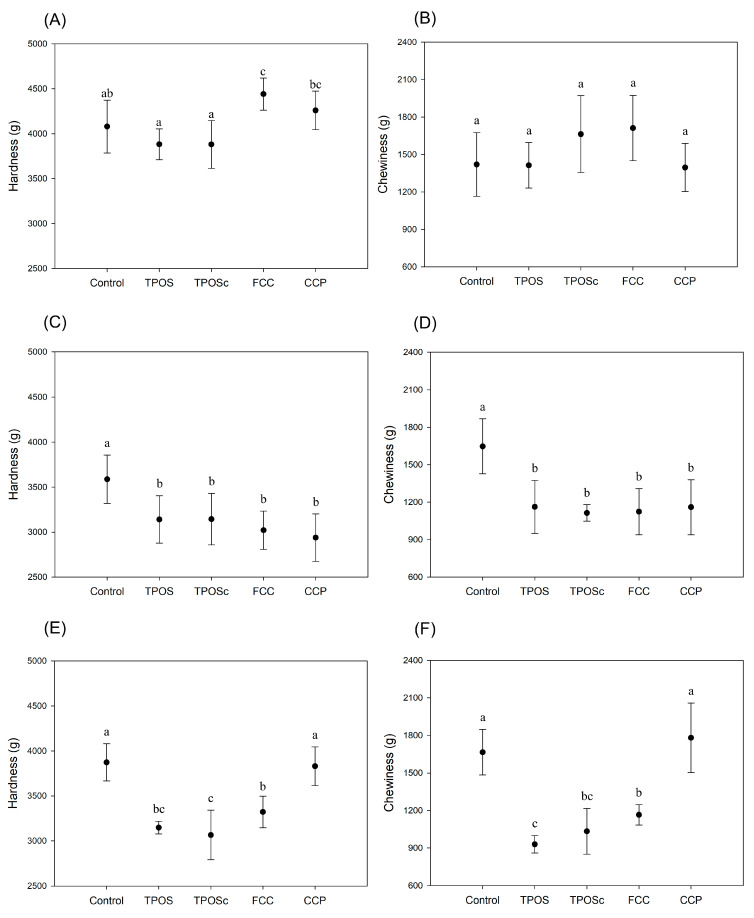
Hardness and chewiness of rice cakes supplemented with calcium compounds at (**A**,**B**) 0.1 wt%, (**C**,**D**) 0.3 wt%, and (**E**,**F**) 0.5 wt% concentrations. Different letters indicate significant differences among groups according to Duncan’s test (*p* < 0.05).

**Table 1 foods-14-02579-t001:** Size and size distribution.

	Count Rate (kcps)	Z-Average (d.nm)	PDI	Intercept	Size (d.nm)	St Dev (d.nm)
TPOS	141.91	3635.5 ± 635 ^a^	0.557 ± 0.147 ^a^	1.049	946.45 ± 384 ^ab^	111.77 ± 61.6 ^ab^
TPOSc	302.41	1408.1 ± 513 ^b^	0.972 ± 0.0673 ^b^	0.740	1175.3 ± 479 ^b^	180.50 ± 80.1 ^b^
FCC	219.08	1344.3 ± 233 ^b^	0.964 ± 0.0509 ^b^	1.075	476.26 ± 75.4 ^a^	48.910 ± 15.2 ^a^
CCP	327.36	1961.5 ± 646 ^ab^	0.246 ± 0.197 ^c^	0.720	1981.3 ± 618 ^c^	335.38 ± 119 ^c^

Values are expressed as mean ± standard deviation (*n* = 6). Different letters in the same column indicate significant differences using Duncan’s test (*p* < 0.05), whereas the same letters in the same column indicate statistically insignificant differences (*p* > 0.05). PDI, polydispersity index.

## Data Availability

The original contributions presented in this study are included in the article. Further inquiries can be directed to the corresponding author.

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
