# Peer review of "Antimicrobial Properties of Thermally Processed Oyster Shell Powder for Use as Calcium Supplement"

_foods, 2025, doi:10.3390/foods14152579_

Round 1
Reviewer 1 Report (Previous Reviewer 1)
Comments and Suggestions for Authors
here remains a fundamental question as to whether 0.1% or so solubility is sufficient for rendering a desirable effect. Table 1 does not provide number of replicate and standard deviation. Title needs a change as "for calcium supplier" is inappropriate. Results are on samples from 2015-2017; do these remain the same after almost a decade?
Fundamentally, this is not a valuable contribution and content has already been patented, so why to attempt to yet publish again? I suggest rejection although have marked a major revision in case a better justification, additional data and discussion could be provided.
Author Response
Here remains a fundamental question as to whether 0.1% or so solubility is sufficient for rendering a desirable effect.
Response 1
Thank you for your insightful comment regarding the solubility level of our samples and whether it is sufficient to elicit a meaningful effect. We agree that the solubility values observed in our study (approximately 0.5 mg/g for TPOS and 0.4 mg/g for TPOSc) are relatively low compared to highly soluble calcium compounds.
However, despite the modest solubility, antimicrobial tests in both controlled broth and food matrices demonstrated statistically significant antimicrobial effects compared to the control groups. This suggests that factors beyond solubility, such as surface properties and particle size-related interactions, may also contribute to the observed antimicrobial activity.
That said, from a nutritional standpoint, we acknowledge that the calcium delivery remains limited. Based on the solubility data, a 100 g serving of rice cake containing the calcium powders would provide approximately 7.5% of the daily recommended calcium intake highlighting the need to further improve bioavailability for dietary purposes.
Comments 2
Table 1 does not provide number of replicate and standard deviation.
Response 2
Thank you for your valuable comment. In response, we have revised Table 1 to explicitly present the standard deviations for each value. Furthermore, we have added a note below the table indicating that the data were obtained from six independent measurements (n = 6), to clarify the number of replicates used for statistical analysis.
The updated text can be found on page 6-7, subsection 3.1.1. (Table 1)
Comments 3
Title needs a change as "for calcium supplier" is inappropriate.
Response 3
Thank you for your helpful comment regarding the manuscript title. We agree that the term “supplier” may have been inappropriate or potentially ambiguous in this context. Accordingly, we have revised the title to: “Antimicrobial properties of thermally processed oyster shell powder for calcium supplement” to more clearly reflect the purpose and scope of the study.
Comments 4
Results are on samples from 2015-2017; do these remain the same after almost a decade?
Response 4
Thank you for your thoughtful question regarding the age and representativeness of the samples. We appreciate your concern about whether the oyster shells used in this study, which were collected between 2015 and 2017, would still yield comparable results if more recently harvested shells were used especially considering potential environmental changes over time.
We would like to clarify that the key functional components responsible for antimicrobial activity in this study—primarily calcium carbonate (CaCO₃) [1], and its thermally converted products CaO and Ca(OH)â‚‚—are generally consistent across oyster shells regardless of harvest year, particularly when sourced from the same geographical region. Several previous studies have reported that the mineral composition of oyster shells shows little variation across different harvest periods and locations, as the primary component, calcium carbonate, remains stable in mature shells under natural conditions [2,3].
Furthermore, the high-temperature thermal treatment (above 700 °C) used in this study is known to effectively decompose calcium carbonate and eliminate organic impurities. This process reduces the influence of any minor environmental or biological differences in raw shells, leading to a more uniform and predictable material with well-documented physicochemical properties [4]. Therefore, we believe the experimental conditions described in our study would produce comparable results even with more recently sourced oyster shells, assuming similar thermal processing parameters are applied.
Comments 5
Fundamentally, this is not a valuable contribution and content has already been patented, so why to attempt to yet publish again?
Response 5
We sincerely thank the reviewer for raising this important point regarding the relationship between our study and the previously granted patent. We would like to respectfully clarify that the scope of the patent pertains exclusively to the production method of calcium powders derived from oyster shells.
In contrast, the current manuscript is distinctly focused on the functional evaluation of the patented materials particularly their antimicrobial properties and physicochemical behaviors. Furthermore, we explored whether these properties are retained and effective within actual food systems, by applying the materials to rice cakes. While the patented process describes how the calcium powders are produced, this study investigates how those materials behave in practice especially their solubility, pH behavior, morphological changes, and antimicrobial activity when used as food additives.
Therefore, rather than duplicating the content of the patent, this manuscript aims to provide meaningful scientific data on the practical implications and potential of the patented materials. We hope this distinction helps clarify the novelty and contribution of the present work.
- Ulagesan, S.; Krishnan, S.; Nam, T.-J.; Choi, Y.-H. A review of bioactive compounds in oyster shell and tissues. Frontiers in Bioengineering and Biotechnology 2022, 10, 913839.
- Mouchi, V.; De Rafélis, M.; Lartaud, F.; Fialin, M.; Verrecchia, E. Chemical labelling of oyster shells used for time-calibrated high-resolution Mg/Ca ratios: A tool for estimation of past seasonal temperature variations. Palaeogeography, palaeoclimatology, palaeoecology 2013, 373, 66-74.
- Mouchi, V.; Godbillot, C.; Forest, V.; Ulianov, A.; Lartaud, F.; de Rafélis, M.; Emmanuel, L.; Verrecchia, E.P. Rare earth elements in oyster shells: provenance discrimination and potential vital effects. Biogeosciences 2020, 17, 2205-2217, doi:10.5194/bg-17-2205-2020.
- Sadeghi, K.; Park, K.; Seo, J. Oyster shell disposal: potential as a novel ecofriendly antimicrobial agent for packaging: a mini review. Korean J. Packag. Sci. Technol 2019, 25, 57-62.
Reviewer 2 Report (Previous Reviewer 2)
Comments and Suggestions for Authors
I would like to thank the authors for carefully addressing all the comments and suggestions provided in the previous round of review. The revised manuscript reflects these improvements, particularly in terms of clarity, organization, and terminology. Based on the corrections made, I consider that the manuscript now meets the necessary standards for publication. I congratulate the authors on their work and contribution.
Author Response
Thank you very much for your encouraging comments. Your thoughtful suggestions were instrumental in improving the manuscript, and we sincerely appreciate your support throughout the review process.
Reviewer 3 Report (Previous Reviewer 3)
Comments and Suggestions for Authors
My previous comments to the authors are addressed. Still, the authors should consider correcting the following existing mistakes in the newly revised sections:
- In subsections 2.2 and 3.1, the authors used the word “Physicochemistic”, which is inaccurate presented this word as “physicochemical” or “physiological” in previous rounds of peer-review. These two subsections cover both analytical protocols for material characterization and biological tests. Therefore, please simply modify the heading as “Analytical Methods”.
- In lines 255-256, the sentence: “These findings aligned with the corresponded well with the sensory evaluation results described earlier in this section.”
The sentence is incomplete and should be rewritten.
Comments on the Quality of English LanguageSentence structure still requires improvement (please see my enclosed comment).
Author Response
Reviewer 3
Comments 1
In subsections 2.2 and 3.1, the authors used the word “Physicochemistic”, which is inaccurate presented this word as “physicochemical” or “physiological” in previous rounds of peer-review. These two subsections cover both analytical protocols for material characterization and biological tests. Therefore, please simply modify the heading as “Analytical Methods”.
Response 1
We sincerely apologize for overlooking the incorrect use of the term “Physicochemistic” in the previously revised version. Given that Sections 2.2 and 3.1 cover both material characterization and biological testing, we believe that the term “Analytical” more accurately reflects the content and scope of these sections. Thank you for your careful attention and valuable suggestion.
The updated text can be found on page 3, 6
Subsection 2.2: Analytical Methods
Subsection 3.1: Analytical Results
Comments 2
In lines 255-256, the sentence: “These findings aligned with the corresponded well with the sensory evaluation results described earlier in this section.”
The sentence is incomplete and should be rewritten.
Response 2
Thank you for your thoughtful comment. We have reviewed the sentence and confirmed that “aligned with the corresponded well” was a redundant and grammatically incorrect expression.
The updated text can be found on page 6, subsection 3.1.1.
Round 2
Reviewer 1 Report (Previous Reviewer 1)
Comments and Suggestions for Authors
I have already noted to proceed with its acceptance.
This manuscript is a resubmission of an earlier submission. The following is a list of the peer review reports and author responses from that submission.
Round 1
Reviewer 1 Report
Comments and Suggestions for Authors
The manuscript is on a topic of interest. However, there is a fundamental question regarding the solubility that is so low at less than 0.1% and with 0.3% activity in a g gram rice cake, the question remains as to how many rice cakes one should consume to provide relevant amount of calcium. Results should provide data on daily requirement and what percent of it will be provided by the formulations. There are too many figures and text that could be removed and proper focus be given to what might be really important and novel, if any.
Author Response
Comments 1
The manuscript is on a topic of interest. However, there is a fundamental question regarding the solubility that is so low at less than 0.1% and with 0.3% activity in a g gram rice cake, the question remains as to how many rice cakes one should consume to provide relevant amount of calcium. Results should provide data on daily requirement and what percent of it will be provided by the formulations.
Response 1
Thank you for your comment and suggestion. We agree that in nutritional studies, understanding calcium intake levels relative to daily requirements is important. We wish to clarify that the primary objective of this study is not to evaluate the nutritional value or calcium intake potential of the rice cakes, but rather investigate the antimicrobial properties of oyster shell-derived calcium powders. These materials were incorporated into rice cakes as model food matrices to assess whether their antimicrobial effects could be retained in practical food systems. Accordingly, factors such as solubility and overall calcium content were interpreted in the context of microbial inhibition, rather than dietary fortification or calcium absorption.
Comments 2
There are too many figures and text that could be removed and proper focus be given to what might be really important and novel, if any.
Response 2
Thank you for your suggestions. Accordingly, we streamlined the figures and text for clarity and impact. The XRD results, originally presented in four separate figures (Figures 9–12), were consolidated into a single figure. This reduces redundancy and improves visual clarity while maintaining the scientific integrity of the data. Additionally, the conclusion section has been significantly revised to be more concise and focused. Rather than summarizing all the experimental results, the updated version highlights only the key findings and emphasizes the practical significance and novelty of oyster shell-derived calcium powders as antimicrobial agents in food systems.
We have revised the text on pages 14–26 subsection 3.1.5 (Figure 9) and section 4.
Reviewer 2 Report
Comments and Suggestions for Authors
The manuscript presents an interesting and valuable study that explores the physicochemical modification of oyster shell powder through thermal and chemical treatments, and evaluates its potential as an antimicrobial additive in food systems. The experimental work is thorough, and the analytical techniques used are suitable for the study's objectives. The investigation is relevant from both a scientific and environmental standpoint, as it proposes a potential valorization route for shellfish waste.
The main strengths of the Manuscript are:
- The introduction effectively defines the problem and justifies the use of oyster shells as a sustainable resource.
- The methodology is comprehensive and generally well described, and the experiments are logically designed to assess the physical, chemical, and antimicrobial properties of the powders.
- The results section presents a wide range of valuable data that supports the antimicrobial potential of the materials developed.
Points to be Revised:
- In section 2.2 (line 104), the title “Physiological characteristics” is incorrect. Since the section refers to the analysis of physicochemical properties of the prepared powders, it should be changed to “Physicochemical characteristics”. The same correction applies to section 3.1 (line 200), which also incorrectly uses “Physiological characteristics”.
- The antimicrobial activity against bacteria, currently included in section 2.6, should not be part of the physicochemical characterization. It would be more appropriate to move this content either to section 2.7 (if conceptually integrated) or create a separate section to clearly distinguish it from the physical and chemical analyses.
- The statement in lines 278–280 is unclear: “Based on these results, we further hypothesized that the same experiment was conducted using 50% acetic acid... were monitored over time (Figure 3)”. Please clarify whether the experiment was conducted or if it remains a proposed hypothesis. The sentence should be revised to reflect the intent and action.
- In Figure 15, the initial microbial counts vary considerably across treatments, which makes it difficult to accurately compare the antimicrobial effects. It is recommended that the data be normalized to the same initial microbial load across all treatments to enable a fair comparison of antimicrobial efficacy
- The conclusion is currently too lengthy and serves more as a summary of results. It would be more effective if restructured to succinctly highlight the key findings and emphasize the relevance and potential of oyster shell-derived materials as antimicrobial additives in food applications. This would better align the conclusion with the study’s implications and practical applications.
I suggest that the manuscript undergo a thorough grammar and language review to correct some expression issues and improve the clarity of certain ideas. Additionally, there are some punctuation errors, such as misplaced periods and repeated colons, that should be revised.
Author Response
Comments 1
In section 2.2 (line 104), the title “Physiological characteristics” is incorrect. Since the section refers to the analysis of physicochemical properties of the prepared powders, it should be changed to “Physicochemical characteristics”. The same correction applies to section 3.1 (line 200), which also incorrectly uses “Physiological characteristics”.
Response 1
Thank you for your suggestion. Accordingly, we have revised the section titles in Section 2.2 (line 106) and Section 3.1 (line 214) from “Physiological characteristics” to “Physicochemical characteristics” to improve the clarity and precision of the terminology.
Comments 2
The antimicrobial activity against bacteria, currently included in section 2.6, should not be part of the physicochemical characterization. It would be more appropriate to move this content either to section 2.7 (if conceptually integrated) or create a separate section to clearly distinguish it from the physical and chemical analyses.
Response 2
Thank you for your comment. We recognize the confusing manuscript structure. Although the antibacterial activity (Section 2.6) was intended as a separate section, the preceding sections (2.3–2.5), which also describe physicochemical characteristics such as pH, SEM-EDS, and XRD, were not grouped under Section 2.2. This may have created the impression that Section 2.6 was part of the physicochemical analysis. We have revised the structure and clarified in the manuscript that Sections 2.2.3 to 2.2.5 are part of the overall physicochemical characterization.
We have revised the text on pages 3 and 4 and subsections 2.2.3– 2.2.5.
Comments 3
The statement in lines 278–280 is unclear: “Based on these results, we further hypothesized that the same experiment was conducted using 50% acetic acid... were monitored over time (Figure 3)”. Please clarify whether the experiment was conducted or if it remains a proposed hypothesis. The sentence should be revised to reflect the intent and action.
Response 3
Thank you for your comment. We acknowledge that the original sentence was misleading, implying a hypothetical situation in which an experiment using 50% acetic acid was conducted. The purpose of this experiment was not to evaluate the acid-neutralizing capacity per se, but observe how the calcium-based samples affect the pH in an acidic environment, as a comparative extension of the DI water results shown in Figure 2.
We have revised the text on page 8 and subsection 3.1.3.
"Based on these results, a follow-up experiment was conducted using 50% acetic acid (initial pH 1.4) instead of distilled water. The changes in pH changes monitored over time after adding the calcium-based samples to determine how the pH responded to the acidic environment (Figure 3)."
Comments 4
In Figure 15, the initial microbial counts vary considerably across treatments, which makes it difficult to accurately compare the antimicrobial effects. It is recommended that the data be normalized to the same initial microbial load across all treatments to enable a fair comparison of antimicrobial efficacy.
Response 4
Thank you for your comment. We agree that the original title of Figure 12 suggests that Escherichia coli was deliberately inoculated into the rice cake samples. As described in Section 2.4.4 ‘Antimicrobial activity test,’ this experiment was conducted to observe the natural microbial growth of the samples stored at room temperature (25 °C), without artificial inoculation. Since the microbial load was not standardized, and no specific strains were introduced, the initial microbial counts may vary across samples based on naturally occurring contamination during preparation. To avoid confusion, we revised the figure title to indicate that the results represent changes in naturally occurring microbial populations under ambient storage conditions. Accordingly, when interpreting the antimicrobial effects of the calcium-based samples, we focused on qualitative trends, such as the degree of microbial suppression or delay in growth observed over time for each treatment, not on the absolute microbial counts but rather.
We have revised the text on page 23, Figure 12.
Comments 5
The conclusion is currently too lengthy and serves more as a summary of results. It would be more effective if restructured to succinctly highlight the key findings and emphasize the relevance and potential of oyster shell-derived materials as antimicrobial additives in food applications. This would better align the conclusion with the study’s implications and practical applications.
Response 5
Thank you for your comment. Accordingly, we revised this section to succinctly present the main findings and highlight the practical relevance and potential of oyster shell-derived antimicrobial materials. auth-info
We have revised the text on page 26 and section 4.
"This study developed and evaluated antimicrobial agents derived from oyster shell-based calcium powders, namely, TPOS, TPOSc, FCC, and CCP, by investigating their physicochemical characteristics and effectiveness in controlled environments and real food matrices, such as starch-based rice cakes. Antimicrobial activity tests showed that TPOSc had meaningful antimicrobial effects at concentrations above 0.5%, indicating that ion release and physical characteristics, besides alkalinity, contributed to antimicrobial performance. Calcium-based materials were incorporated into starch-based rice cakes to evaluate their applicability to actual food systems. FCC and TPOS showed stronger antibacterial and antifungal activities than TPOSc at concentrations > 0.3 wt%, confirming that their antimicrobial functionality was retained within real food matrices. These findings introduce a naturally derived antimicrobial agent suitable for food applications and provide a more nuanced understanding of how factors, such as ion release, solubility, surface characteristics, and alkalinity, contribute to antimicrobial efficacy. While this study demonstrated the clear antimicrobial potential of oyster shell-derived materials in a rice cake model system, its broader utility remains to be explored. Expanding this approach to other food systems could validate the versatility and sustainability of these materials. Finally, this study will aid in transforming marine biowaste into value-added functional food-preserving agents with practical industrial relevance."
Reviewer 3 Report
Comments and Suggestions for Authors
The manuscript discusses the impact of thermal processing of oyster shell waste for improving the antioxidant and antimicrobial properties in rice cakes. The topic of the work should be of interest to the journal’s readership. However, the manuscript must be improved by an inclusion of additional information in some sections. The authors should also pay a specific attention to accurate presentation of the statistical analysis results in the figures.
I have the following comments to the authors:
- In line 43, bring examples of environmental challenges associated with calcium carbon-rich-shell
- In section 2.7.5, add additional details on how hardness and chewiness were measured and provide an adequate reference.
- In line 217, “aligned with the corresponded well”, this part of the sentence should be rewritten.
- In lines 245-246, why citric acid-treated oyster shell showed lower solubility?
- In lines 246-247, Add a brief reason to the main text on why FCC had the highest solubility among all samples.
- The results of the statistical analysis and Duncan's test (letters above the bars) are missing in Figure 1 and this figure should be updated.
- In lines 480-481, the sentence: “These characteristics…calcium-based samples”
The sentence is incomplete and should be rewritten.
- In line 63, the reference “Yen et al. 2022” should be provided in a numerical format.
Comments on the Quality of English Language
English language should be improved (please see my comments)
Author Response
Comments 1
In line 43, bring examples of environmental challenges associated with calcium carbon-rich-shell
Response 1
Thank you for your comment and suggestion. Accordingly, we have revised the text in line 43 to provide a clearer context for the environmental issues associated with waste shells. Specifically, we indicated that waste shells cause problems such as strong odor during microbial decomposition and heavy metal leaching during weathering processes.
We have revised the text on page 1, subsection 1.
"The calcium carbonate (CaCO₃)-rich shell matrix, which constitutes most of the shell, remains largely underutilized. Waste shells exert adverse environmental effects, such as strong odor generation during microbial decomposition and heavy metal leaching from shell piles exposed to weathering [3]."
Comments 2
In section 2.7.5, add additional details on how hardness and chewiness were measured and provide an adequate reference.
Response 2
Thank you for your comments. Accordingly, we have revised Section 2.7.5 to include additional details on how hardness and chewiness were measured using the Texture Profile Analysis (TPA) method. Specifically, we added information on the pre-test, test, and post-test speeds, strain percentage, trigger force, and sample dimensions. These measurements were conducted using a TA-XT texture analyzer equipped with a 5 kg load cell, following standardized procedures commonly used for starch-based food matrices. Additionally, we included relevant references to support the methodology.
The updated text can be found on page 5, subsection 2.4.5.
"The texture of the rice cakes was analyzed using a texture analyzer (TA-XT, MHK Corp., Anyang, Gyeonggi, Korea) equipped with a 5 kg load cell. The samples were prepared in cylindrical shapes with a diameter of 1.2 cm and a height of 1 cm. Hardness and chewiness were evaluated using the Texture Profile Analysis (TPA) method. The test was conducted using a special test mode with the following conditions: pre-test speed of 1.0 mm/s, test speed of 1.0 mm/s, and post-test speed of 2.0 mm/s. The strain was set at 85%, and the trigger force was set at 5 g. Measurements were taken in triplicate, and the results were analyzed to determine hardness and chewiness. This method was adapted from previous studies that used similar starch-based food matrices [1,2]."
Comments 3
In line 217, “aligned with the corresponded well”, this part of the sentence should be rewritten.
Response 3
Thank you for your suggestion. Accordingly, we have revised the text on page 6, subsection 3.1.1.
"These findings are consistent with the sensory evaluation results described earlier in this section."
Comments 4
In lines 245-246, why citric acid-treated oyster shell showed lower solubility?
Response 4
Thank you for your comment. Although our study did not specifically investigate the reason for the slightly lower solubility of citric acid-treated oyster shells (TPOSc) than that of TPOS, we agree that the previous wording might have unintentionally emphasized this difference. To clarify this, we revised the text to present the solubility values as part of a general trend without implying a specific interpretation of the difference.
We revised the text on page 7, subsection 3.1.2.
"As shown in Figure 1, the measured solubilities of the reference compounds closely matched their known solubilities, confirming the validity of the experimental method. TPOS and TPOSc exhibited a solubility of approximately 0.5 and 0.4 mg/g, respectively, significantly higher than pure CaCO₃ but approximately one-third that of Ca(OH)2."
Comments 5
In lines 246-247, Add a brief reason to the main text on why FCC had the highest solubility among all samples.
Response 5
Thank you for your suggestion. We have added a brief explanation to the main text. We have revised the text on page 7, subsection 3.1.2.
"This high solubility is likely attributable to the partial conversion of FCC to reactive CaO during calcination, along with its fibrous morphology, which increases the surface area and facilitates more efficient dissolution in aqueous environments.'
Comments 6
The results of the statistical analysis and Duncan's test (letters above the bars) are missing in Figure 1 and this figure should be updated.
Response 6
Thank you for your comments. Accordingly, we revised Figure 1 to include the results of the statistical analysis, specifically letters representing Duncan’s multiple range test placed above the bars to indicate statistically significant differences among the samples.
We revised the text on page 7, subsection 3.1.2.
Comments 7
In lines 480-481, the sentence: “These characteristics…calcium-based samples”. The sentence is incomplete and should be rewritten.
Response 7
Thank you for your comment. We have revised the text to better convey the rationale behind the selection of the rice cake and its relevance to the study's objectives.
We revised the text on page 19, subsection 3.3.1.
"Due to its high susceptibility to microbial spoilage and simple composition, rice cake was selected as a suitable model food matrix to evaluate the effects of calcium-based samples on pH, discoloration, and fungal contamination under ambient storage conditions."
Comments 8
In line 63, the reference “Yen et al. 2022” should be provided in a numerical format.
Response 8
Thank you for your suggestion. We have revised the manuscript accordingly to ensure all references comply with the journal’s citation style.
We revised the text on page 2, subsection 1.
Round 2
Reviewer 1 Report
Comments and Suggestions for Authors
The work lacks concentration on the antimicrobial effects that the authors now claim to be the main purpose. However, there is no background or logic as to why this was expected in the first place. In addition, the bulk of the work is on matters unrelated to what the author mention was the main purpose. No information is provided on where and when the samples were collected and why only one batch of sample was used. There are a multitude of questions that make me now reject the manuscript for further consideration.
Reviewer 3 Report
Comments and Suggestions for Authors
My previous comments are addressed and I have no additional comment to the authors.